# Development of an Anti-HER2 Single-Chain Variable Antibody Fragment Construct for High-Yield Soluble Expression in *Escherichia coli* and One-Step Chromatographic Purification

**DOI:** 10.3390/biom13101508

**Published:** 2023-10-11

**Authors:** Kyu Tae Byun, Boram Kim, Junmin Cho, Inbeom Lee, Myung Gu Lee, Dongsun Park, Tae-Bong Kang, Hyung-Sik Won, Chan Gil Kim

**Affiliations:** 1Department of Biotechnology, Research Institute (RIBHS), College of Biomedical and Health Science, Konkuk University, Chungju 27478, Republic of Korea; bgt1002@kku.ac.kr (K.T.B.); kbr9492@kku.ac.kr (B.K.);; 2Konkukbio Inc., Konkuk University, Chungju 27478, Republic of Korea; 3Department of Biology Education, Korea National University of Education, Cheongju 28173, Republic of Korea; 4BK21 Project Team, Department of Applied Life Science, Graduate School, Konkuk University, Chungju 27478, Republic of Korea

**Keywords:** antibody, single-chain variable fragment (scFv), human epidermal growth factor receptor 2 (HER2), bacterial production, *Escherichia coli*, maltose-binding protein (MBP), tobacco etch virus (TEV) protease

## Abstract

Although single-chain variable fragment (scFv) is recognized as a highly versatile scaffold of recombinant antibody fragment molecules, its overexpression in *Escherichia coli* often leads to the formation of inclusion bodies. To address this issue, we devised and tested four different constructs, named v21, v22, v23 and v24, for producing anti-human epidermal growth factor receptor 2 (HER2) scFv. Among them, the v24 construct obtained from N-terminal fusion of maltose-binding protein (MBP) and subsequent tobacco etch virus protease (TEV) was identified as the most efficient construct for the production of anti-HER2 scFv. Aided by an MBP tag, high-yield soluble expression was ensured and soluble scFv was liberated in cells via autonomous proteolytic cleavage by endogenously expressed TEV. The isolated scFv containing a C-terminal hexahistidine tag was purified through a one-step purification via nickel-affinity chromatography. The purified scFv exhibited a strong (nanomolar *K*_d_) affinity to HER2 both in vitro and in cells. Structural and functional stabilities of the scFv during storage for more than one month were also assured. Given the great utility of anti-HER2 scFv as a basic platform for developing therapeutic and diagnostic agents for cancers, the v24 construct and methods presented in this study are expected to provide a better manufacturing system for producing anti-HER2 scFv with various industrial applications.

## 1. Introduction

Since hybridoma technology was developed in the mid-1970s, immunoglobulin G (IgG)-based monoclonal antibodies (mAbs) have served as an invaluable asset for developing antibody-based therapeutic and diagnostic agents for various diseases. Advances in recombinant protein technology have led to rapid progress in the application of antibody engineering, including antibody-drug conjugates (ADCs), bispecific antibodies (biAbs), and diverse scaffolds of antibody fragments (AbFs) [1,2,3,4]. In particular, recombinant AbFs have emerged as alternative antibody scaffolds that are highly versatile and amenable to advanced engineering for ADC, biAb, and diagnostic applications. Single-chain variable fragment (scFv) represents the smallest of the Ig-based AbF scaffolds capable of antigen binding. It is composed of variable domains of light chain and heavy chain (V_L_ and V_H_, respectively) covalently linked via a short peptide linker. Although scFv is also one of the most classical AbF scaffolds, it is appreciated as a highly versatile molecule that has led antibody technology in both fields of academic research and clinical applications. For instance, the scFv scaffold has long been utilized for diverse biological display technologies to identify ligand-receptor pairs, including covalent DNA display, mRNA display, ribosomal display, virus/phage display, and cell display [5,6]. In particular, the most popular phage-display technology using scFv has greatly contributed to the development of novel antibody molecules. Many of the state-of-the-art immunotherapy technologies such as bispecific T-cell engagers and chimeric antigen receptor T-cell therapy have also been developed using scFv [7,8].

As blockbuster biopharmaceutical therapeutic agents, trastuzumab (Herceptin^®^, Roche, Basel, Switzerland), a humanized mAb, and its ADC variant, ado-trastuzumab emtansine (Kadcyla^®^, Roche, Basel, Switzerland), are frequently administered to treat breast cancer [9,10]. Trastuzumab specifically targets the extracellular domain of human epidermal growth factor receptor 2 (HER2), a tumor surface antigen overexpressed in approximately 15–30% of cases of human breast cancer [11]. Several anti-HER2 scFv-based agents are also under development as therapeutic or diagnostic alternatives to trastuzumab [10,12]. As scFv consists of minimalistic domains accomplishing antigen binding, benefits of higher target selectivity, reduced immunogenicity, or both are expected when using scFv as an alternative to the whole antibody. Possible bacterial production is also an attractive merit of using AbFs. Compared to whole antibody production, which usually requires a costly system based on mammalian cell cultures to enable appropriate glycosylation, the bacterial system is amenable to the production of AbFs including scFvs [13], which are usually devoid of any post-translational modifications. Besides the advantage of using low-cost facilities, bacterial production of recombinant proteins with *Escherichia coli* (*E. coli*) as the most popular host has numerous advantages, including straightforward maintenance and manipulation of cells, simple optimization of process, and high-level expression [14,15]. However, cytoplasmic expression of scFvs in *E. coli* often results in the formation of inclusion bodies, which are insoluble aggregates of expressed proteins with non-native, and thus non-bioactive, structures. To promote the soluble expression of proteins of interest, several approaches have been attempted including fusion of soluble tag protein [15,16], co-expression of molecular chaperones and folding modulators [17,18], modification of media composition for secretion [19,20], refolding using detergents and additives [21], and expression in different host systems [22]. Among them, expression with soluble fusion tag protein is often employed as the default option to improve the solubility of target proteins. However, with this approach, an additional process of enzymatic cleavage is required to remove the tag protein from the target protein of interest. In addition, cleavage of the tag protein after purification of the whole fusion protein often interferes with the folding stability of the remaining target protein, making it more difficult to perform functional assays [16]. In this study, to address those concerns (enhancing solubility and facilitating process) in anti-HER2 scFv production in *E. coli*, we designed and assessed four different constructs, named v21, v22, v23 and v24, expressing recombinant scFv. Results suggested that the v24 construct was the most efficient production system for anti-HER2 scFv in *E. coli* with enhanced solubility and straightforward purification. Structure functional validation of the final product generated using the v24 construct was then performed.

## 2. Materials and Methods

### 2.1. Materials

To construct the v21 to v24 recombinant plasmids, DNA fragments encoding the anti-HER2 scFv tagged with an N-terminal TEV cleavage site and a C-terminal Myc-hexahistidine (H_6_) fusion tag (t-scFv-MH) were chemically synthesized by Cosmogentech (Seoul, Republic of Korea). MDA-MB-231 human breast cancer cell line and SKOV3 human ovarian cancer cell line were obtained from the American Type Culture Collection (Manassas, VA, USA) and maintained by culturing at 37 °C with 5% CO_2_ in a high-glucose Dulbecco’s modified Eagle medium or Roswell Park Memorial Institute 1640 medium supplemented with 10% fetal bovine serum, penicillin (100 U/mL), and streptomycin (100 μg/mL). *E. coli* DH5α and BL21(DE3) strains used for gene subcloning and protein expression studies, respectively, were cultured in Luria-Bertani (LB) medium consisting of 10 g/L tryptone, 5 g/L yeast extract, 10 g/L NaCl, and 100 μg/mL ampicillin. Vector plasmids pRK793 and pCold III were obtained from David Waugh (Addgene plasmid #8827, Stoney Stanton, UK) and Takara Bio Inc (#3363, Shiga, Japan), respectively. An Ni-NTA column for nickel-affinity chromatography was purchased from Thermo Fisher Scientific (#88222, Waltham, MA, USA). MBPTrap HP and Capto-L columns for amylose-affinity and antibody-affinity chromatography, respectively, were purchased from GE Healthcare (#28-9187-79 and #17-5478-01, respectively, Chicago, IL, USA). Chromatographic procedures using these columns were performed according to the manufacturer’s protocols. Protein-L-HRP used for enzyme-linked immunosorbent assay (ELISA) and Protein-L-FITC used for fluorescence-activated cell sorting (FACS) analysis were purchased from Genscript (#M00098, New Jersey, NJ, USA) and Acro biosystem (#RPL-PF141, Newark, DE, USA), respectively. The extracellular domain of HER2 (HER2-ExD) for ELISA was also purchased from Acro Biosystems (#HE2-H5225, Newark, DE, USA). Tobacco etch virus protease (TEV) was prepared as a recombinant protein using the pRK793 vector plasmid, as described previously [15,23].

### 2.2. DNA Recombination

Figure 1 describes the compositions of the v21 to v24 recombinant plasmids. To prepare the v23 construct, chemically synthesized, codon-optimized t-scFv-MH oligonucleotides were inserted between the *Sac*I and *Pst*I restriction sites of a pRK793 vector plasmid. For generation of the v21 construct, the region from V_L_ to H_6_ in the v23 plasmid was PCR-amplified. The PCR products were then restricted, followed by ligation between the *Nde*I and *Pst*I restriction sites of pCold-III vector. The v22 plasmid was constructed by inserting PelB sequence-encoding oligonucleotides at the *Nde*I restriction site of the v21 construct. Finally, to generate the v24 construct, TEV-encoding oligonucleotides, which were PCR-amplified from the pRK793 vector plasmid, were inserted into the v23 plasmid using the *Sac*I restriction site.

### 2.3. Protein Sample Preparation

Recombinant plasmids (v21–v24) were verified by DNA sequencing and subsequently transformed into *E. coli* strain BL21(DE3). A single colony of transformants was inoculated in 3 mL of LB broth medium supplemented with 50 μg/mL of ampicillin and incubated overnight at 37 °C. The culture was scaled up to 50 mL of LB media containing 50 μg/mL of ampicillin and cultivated until OD_600_ reached 0.6–1.0. Protein expression was then induced by adding isopropyl β-_D_-1-thiogalactopyranoside (IPTG) to a final concentration of 0.5 mM. Protein expression was prolonged for 16 h at 15 °C, followed by cell harvest through centrifugation. Collected cell pellets were resuspended in phosphate-buffered saline (PBS) and lysed using sonication. After removing cell debris by centrifugation, expressed proteins were purified from supernatants by application of chromatography using an FPLC system. In brief, amylose-affinity chromatography was performed for maltose-binding protein (MBP)-fused proteins, whereas isolated scFv proteins were purified by antibody-affinity or nickel-affinity chromatography. In vitro enzymatic cleavage of the MBP-scFv fusion protein was performed by treating protein solution (1 μM) with purified TEV (3 μM). The concentration of purified protein in solution was determined spectrophotometrically using molar absorptivity at 280 nm, which was deduced from the amino acid sequence of each protein sample (Appendix A) using the web-based ProtParam tool (https://web.expasy.org/protparam/): 50,100 and 117,940 M^−1^·cm^−1^ for isolated scFv and MBP-scFv fusion protein, respectively.

### 2.4. Enzyme-Linked Immunosorbent Assay

To estimate in vitro specific binding of a protein sample to a specific antigen, ELISA was performed as described previously [15,24]. In brief, the recombinant HER2-ExD antigen (50 ng/mL) or bovine serum albumin (BSA) was used to coat 96-well plates and incubated at 4 °C overnight. After blocking the remaining binding space, protein samples were added to wells at designated concentrations ranging from 0.1 pM to 400 nM, followed by incubation at 37 °C for 2 h. Antigen-specific binding of each sample was then detected by treatment with HRP-conjugated Protein L, followed by enzyme reaction with TMB (Invitrogen, Carlsbad, CA, USA) as a substrate. Color development was stopped with 1 M HCl and the absorbance at 450 nm was measured using a microplate reader. The ELISA immune plate was washed three times between every step of coating, blocking, binding, detection, and reaction procedures. For competitive ELISA, hexahistidine-tagged scFv samples (400 nM, 100 μL) attached to coated HER2-ExD (50 ng/mL) were displaced with increasing concentrations of trastuzumab (Herceptin^®^), a higher-affinity antibody, followed by detection of residual scFv using anti-hexahistidine-HRP.

### 2.5. Flow Cytometry

FACS was used to monitor antigen binding of protein samples on the cell surface. Briefly, cultured (approximately 10^6^) cells (SKOV3 or MDA-MB-231) were incubated with individual protein samples at concentrations ranging from 0.1 pM to 400 nM, for 30 min at 4 °C. These cells were then washed twice with a flow cytometry staining buffer (1% BSA/PBS), followed by a 15 min incubation with 5 μg/mL of an FITC-labeled protein L to detect the bound scFv and trastuzumab. After washing twice with flow cytometry staining buffer, flow cytometric assay of cells was performed using a FACScan instrument (Becton Dickinson, Franklin Lakes, NJ, USA). Quantification of results was conducted using CellQuest software (version 5.2.1, Becton Dickinson, Franklin Lakes, NJ, USA). Additionally, to determine background fluorescence, cells were incubated with an anti-polyhistidine antibody and FITC-labeled anti-mouse IgG without any treatment of scFv.

### 2.6. Statistics

Statistical analyses for ELISA and FACS experiments were performed using GraphPad Prism 5 software (www.graphpad.com). Data were represented as average values with standard deviations (SDs) for three independent experiments. The dissociation constant (*K*_d_) was determined through nonlinear least-squares fitting.

## 3. Results

### 3.1. Design of Anti-HER2 scFv-Expressing Construts

The primary objective of this study was to develop an efficient process of obtaining functional anti-HER2 scFv, including soluble expression (upstream process) and a simple step of purification (downstream process). To achieve this, we designed four types of recombinant plasmid constructs, named v21–v24, carrying anti-HER2 scFv (Figure 1). For the scFv component shared in all constructs, amino acid sequences for variable domains of the light chain (V_L_) and heavy chain (V_H_) covalently linked via a peptide linker sequence (G_4_S)_3_ were derived from the corresponding region in trastuzumab. To ensure efficient expression in *E. coli*, codon-optimized oligonucleotides encoding the scFv sequence were incorporated into vector plasmids. The scFv compartments also commonly carried two C-terminal tags, one Myc tag to facilitate flow-cytometric analysis and a hexahistidine (H_6_) tag for application of nickel-affinity chromatographic purification. For the basic construct, v21, the minimal composition of scFv-Myc-H_6_ was incorporated into pCold III vector plasmid for low-temperature (15 °C) expression known to be beneficial for high-purity soluble expression and increased stability of expressed proteins [25]. In the second construct, v22, an extracellular export sequence, PelB, was additionally attached to the N-terminus of scFv to induce its periplasmic translocation known to help proper folding of antibody fragments [26]. The other two constructs, v23 and v24, were established by modifying pRK793 vector plasmids harboring MBP and TEV fusion tag proteins with a subsequent TEV cleavage site [27]. In the v23, the MBP tag to guide soluble expression of scFv was linked to the scFv via a TEV cleavage sequence for segregation of MBP and scFv by in vitro treatment with TEV after purification of the MBP-scFv fusion protein. Finally, as an extension of v23, the v24 construct contained a TEV tag following the MBP tag to enable isolation of scFv in cells via autocleavage by endogenously expressed TEV.

### 3.2. Comparison in Soluble Expression and Purification

The SDS-PAGE images in Figure 2 summarize the protein expression and purification results of individual constructs. All constructs well overexpressed the designated recombinant proteins with expected molecular sizes (lanes T in Figure 2). In particular, it was confirmed that the intended in-cell cleavage of v24 products occurred upon expression, as it showed two expression bands segregated, one matching the presumable size (approximately 69 kDa) of MBP-TEV fusion protein and the other corresponding to the expected size (approximately 30 kDa) of isolated scFv. After cell lysis by sonication, v21 and v22 products appeared to a great extent in pellet fractions of cell lysates (lane P in Figure 2), suggesting the likely formation of inclusion bodies with non-native structures of proteins. Subsequent purification of scFv from supernatants by nickel-affinity chromatography resulted in very low yields of collection (Table 1) with relatively appreciable inclusion of impurities (lane E in Figure 2).

In contrast, proteins expressed by v23 and v24 were predominantly contained in supernatants of cell lysates (lane S in Figure 2), supporting their soluble expression with proper folding to functional structures. High-purity purification of the MBP-scFv fusion protein expressed by the v23 construct was achieved using an amylose-affinity column (lane E for v23 in Figure 2), which could specifically capture MBP. However, for isolation of scFv, treating the purified MBP-scFv with TEV was necessary to remove the MBP tag by proteolytic cleavage (Figure 3). Following a cleavage reaction for 24 h, the isolated scFv was purified from the reaction mixture by protein-L-affinity chromatography (lane E in Figure 3), which could specifically capture the kappa light chain of antibodies.

In contrast to the v23 product, v24-expressed proteins needed no in vitro process of proteolytic cleavage to remove the MBP tag because the scFv could be autonomously liberated into cells as noted in Figure 2 (lane T for v24). Furthermore, a single chromatographic purification using a nickel-affinity column, which could specifically capture the C-terminal H_6_ tag of the isolated scFv, was sufficient for obtaining high-purity scFv (lane E for v24 in Figure 2). Finally, the productivity of scFv using the v24 construct was found to be higher than that of v23, as it yielded more than 3 mg of purified scFv on a basis of 1000 mg of wet cell pastes (Table 1).

### 3.3. Comparison in Antigen-Binding Activity

ELISA and flow cytometric assay were used to determine antigen-binding activities of v21–v24 product proteins (Figure 4). In direct ELISA with protein-L as a primary antibody for detection, HER2-ExD and BSA were employed as target antigen and negative control antigen, respectively, and trastuzumab (Herceptin^®^) was used as a positive control for antigen binding. ELISA for monitoring in vitro interaction indicated that all scFv samples could recognize HER2-ExD as their specific antigen by showing approximately one nanomolar dissociation constant (*K*_d_) indicative of a strong binding (Figure 4A). FACS results also revealed specific antigen binding of scFvs in cells (Figure 4B).

All four scFv samples showed approximately 10 nM of *K*_d_ values for HER2-overexpressing (HER2-positive) SKOV3 cells, whereas they showed no significant binding for HER2-underexpressing (HER2-negative) MDA-MB-231 cells. In contrast, the initial product of v23, which was an MBP-scFv fusion protein, showed drastically (more than 150 folds) decreased binding affinity to HER2 in ELISA and no measurable binding to SKOV3 cells in FACS analysis. In addition, although antigen-binding affinities of all scFv samples were apparently comparable to one another, the estimated *K*_d_ value of the v24 scFv in both in vitro (ELISA) and cell (FACS) assays appeared to be lower, albeit slightly, than those of other scFvs. Collectively, results indicated that the v24 construct was the most suitable system considering both productivity and product quality. Therefore, the final scFv product of v24 was used for the following identification and validation studies for functionality and stability.

### 3.4. Identification and Validation of Functionality and Stability

As a structural identification, the exact molecular mass of the v24 scFv estimated by mass spectrometry appeared to be 29,627 Da (Figure 5A), which was in good agreement with the theoretical molecular weight of 29,623 Da deduced from its amino acid sequence (Appendix A). Stability of the scFv with regard to structure and activity was inspected for 32 days of storage at 4 °C. There was no significant degradation based on SDS-PAGE analysis (Figure 5B). ELISA results indicated that its antigen-binding activity was completely retained during storage (Figure 5C). Finally, a competitive ELISA was performed to investigate whether the antigen-bound scFv was displaced by a commercial Herceptin^®^ product (Figure 5D). The result indicated dissociation of scFv from the bound antigen by added trastuzumab in a concentration-dependent manner, suggesting that our scFv shared the same epitope as the anti-HER2 mAb, trastuzumab.

## 4. Discussion

ScFv molecules are widely used as a basic platform to develop therapeutic and diagnostic biopharmaceutical agents against various diseases, particularly cancer [28,29]. Considering the necessity and availability of neutralizing HER2 in targeted therapy of many breast cancers, demands are also increasing for establishing an efficient production system of anti-HER2 scFv molecules. However, the most economical production of scFv in *E. coli* is often hindered by its misfolding leading to the formation of inclusion bodies, which inevitably results in loss of activity and reduced productivity in functional scFv. As shown in the results of the v21 construct, it was confirmed that anti-HER2 scFv also formed inclusion bodies upon cytosolic expression in *E. coli*. In addition, as shown with the v22 construct, the use of a signal peptide for extracellular export was ineffective for ensuring proper folding of scFv. Alternatively, for producing recombinant proteins in *E. coli*, using a soluble fusion tag protein is a popular approach to enhance soluble expression of otherwise insoluble proteins. Various fusion tag proteins including bacteriophage T7 protein kinase, small peptide tags, monomeric mutant of the Ocr protein of bacteriophage T7, and glutathione S-transferase (GST) have also been employed to improve the solubility of recombinant proteins to which tags are fused [30,31,32]. It is known that MBP is more effective than other conventional fusion tag proteins such as GST and thioredoxin for both expression level and proper folding of target polypeptides [33,34,35]. Therefore, we chose MBP as the tag protein for our anti-HER2 scFv. Our previous study also found that MBP fusion is superior to fusion of green fluorescence protein for non-peptide guided extracellular secretion in terms of its stabilizing effect on secreted target protein [15]. Consequently, as commonly evidenced by the v23 and v24 constructs in the present study, the use of MBP as a preceding (N-terminal) fusion tag protein was confirmed to be highly effective for ensuring solubility with functional folds of the fused anti-HER2 scFv in *E. coli* cytosolic expression.

In spite of its great utility for soluble expression, potential drawbacks of using fusion tag protein include impaired functionality of the target protein as the fusion tag protein can obstruct proper folding or steric action of the fused target protein. Consistent with previous observations [15], the present result for the initial product (MBP-scFv fusion protein) of v23 also indicated an appreciably attenuated activity of the scFv in state of fusion with MBP. Given that proteolytic cleavage of fusion tag protein is often hindered owing to steric issues [16], the reduced activity of the MBP-fused scFv is likely attributed to steric interference by MBP. Therefore, it is generally necessary to remove the fusion tag protein from the target protein after purification, which is typically achieved by proteolytic cleavage. Fortunately, the present scFv product using the v23 construct remained intact and soluble after cleavage of the MBP fusion tag. However, the tag-removing process is challenging in many cases where the target protein can be truncated by an enzymatic reaction or revert to insoluble aggregates upon the removal of the tag. In addition, the target protein after removal of the tag should be further purified to be separated from the tag protein and proteolytic enzyme used. In this context, an obvious demerit of using fusion tag protein is represented by rendering the whole production process complicated, labor-intensive, and costly [16]. To address this concern, the present v24 construct was used to test the feasibility of autonomous in-cell cleavage of the tag protein, MBP, with an additional fusion of a protease, TEV. The result demonstrated that this clever system operated well as intended by showing efficient liberation of functional scFv in cells without forming inclusion bodies. This result is in sharp contrast to severe aggregation in cells of MBP-untagged scFv produced with the v21 construct. This also indicated that the preceding expression of the MBP tag even with intervention of TEV between MBP and scFv was still effective for ensuring proper folding and solubility of the finally synthesized anti-HER2 scFv. The endogenously isolated v24 scFv could be then purified via a one-step purification process using nickel-affinity chromatography as it was constructed to accompany a C-terminal H_6_ tag. Antibody-affinity chromatography as tested for the v23 scFv can be applied alternatively to nickel-affinity chromatography or for further polishing of the purified scFv. However, the one-step purification by nickel-affinity chromatography is economically desirable. Finally, as shown in Table 1, the v24 construct-based production yield of functional anti-HER2 scFv was more than 21 folds higher than the production yield without using any tag protein (v21 construct) and more than 1.4 folds higher than that without introducing the autonomous tag cleavage in cells (v23 construct). Taken together, these results suggest that the v24 construct was the most optimized system for efficient production of anti-HER2 scFv, although it could not be guaranteed that the v24 system would work for other scFvs or other types of AbFs. Additionally, the final product of v24 showed sufficient stability maintaining structure and activity during storage. The subnanomolar *K*_d_ value (0.89 nM determined by ELISA) of our scFv estimated for the HER2-ExD binding in vitro was superior to those (approximately 100 [36] and 5 nM [37] of *K*_d_ determined by surface plasmon resonance) of other known anti-HER2 scFvs. Even in cells, our scFv exhibited a nanomolar *K*_d_ value (8.45 nM determined by flow cytometry) for its binding to cell-surface-expressed HER2, which was lower (i.e., higher affinity binding) than the previously reported *K*_d_ value of approximately 20 nm [38]. This binding affinity appeared to be approximately three-fold lower than that (2.56 nM of *K*_d_) observed for the parent molecule, trastuzumab (Herceptin^®^). However, considering a more than five-fold higher molecular weight and a divalent nature of the parent molecule, the potency of our small, monovalent Abf (scFv), in terms of effective dose for administration, would be evaluated as comparable to that of the whole antibody, trastuzumab.

In summary, this study generated and compared four different production systems of anti-HER2 scFv in *E. coli* to evaluate the utility of an MBP tag and its autonomous in-cell cleavage. Finally, the v24 construct, which produced the scFv with a retained solubility and activity upon releasing from the MBP tag in cells, was found to be the most efficient production system with the highest productivity and a simple purification process. The final product using this system was validated to represent a strong target-binding affinity to function like a specific antibody. Therefore, we expect that our anti-HER2 scFv would serve as an essential platform leading further developments of novel or improved anti-cancer therapeutic/diagnostic agents, such as ADC and companion diagnostics. In addition, the present strategy and production system hold great promise for efficient and economical large-scale production of various scFvs.

## 5. Patents

An application for a Korean patent (no. 10-2020-0179587) resulted from the work reported in this manuscript.

## Figures and Tables

**Figure 1 biomolecules-13-01508-f001:**
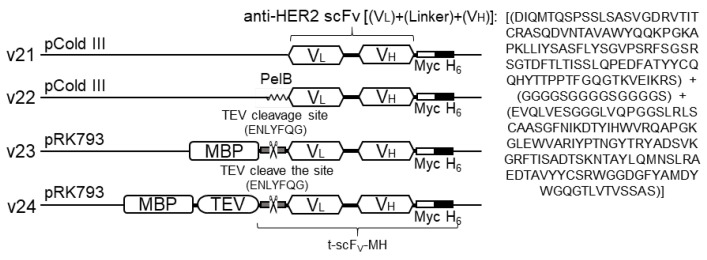
Schematic diagram illustrating the composition of recombinant plasmids (v21–v24) expressing anti-HER2 scFv. The vector plasmid (pCold III or pRK793) used for construction and the locations of fusion tags (MBP, TEV, PelB, Myc, and H_6_) are demonstrated. Amino acid sequences of the TEV cleavage site and scFv are denoted, whereas whole amino acid sequences of recombinant proteins expressed by these four constructs are presented in Appendix A.

**Figure 2 biomolecules-13-01508-f002:**
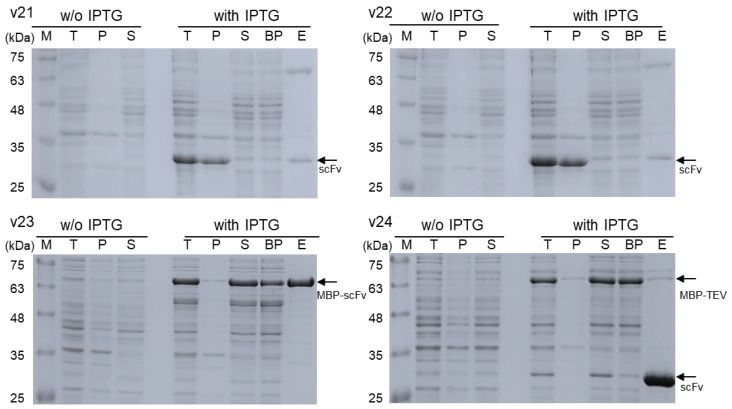
SDS-PAGE images summarizing protein expression and purification results. M, molecular size marker; T, total cell lysates; P, pellet fraction of cell lysates; S, supernatant fraction of cell lysates; BP, pooled solution of binding pass-through fractions from amylose-affinity (v23) or Ni^2+^-affinity (v21, v22, and v 24) chromatography; E, pooled solution of elution fractions in amylose-affinity (v23) or Ni^2+^-affinity (v21, v22, and v 24) chromatography. Positions of individual protein bands are indicated by arrows.

**Figure 3 biomolecules-13-01508-f003:**
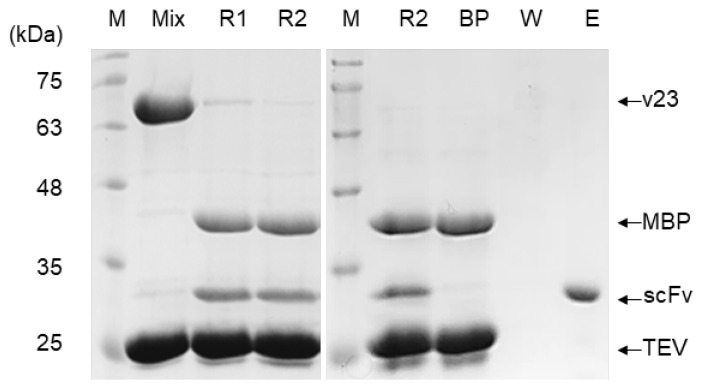
SDS-PAGE images demonstrating the isolation and purification process of the scFv expressed by the v23 construct. Lane M, molecular size marker; Mix, reaction mixture of the v23 product (MBP-scFv fusion protein) and purified TEV; R1 and R2, reaction mixtures after 37 °C incubation for 12 h and 24 h, respectively; BP and W, path-through solutions during binding and washing steps of protein-L-affinity chromatography, respectively; E, final elution fraction from the protein-L-affinity chromatography. Positions of individual protein bands are indicated by arrows.

**Figure 4 biomolecules-13-01508-f004:**
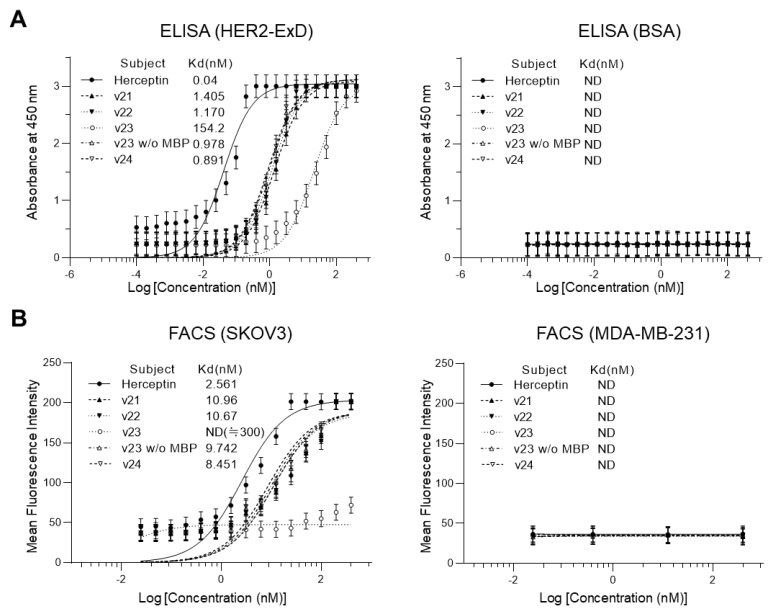
Comparison of antigen-binding activities of v21–v24 products. Data are represented as mean ± SD for triplicate independent experiments. (**A**) ELISA results for monitoring in vitro binding to HER2-ExD (left panel) and BSA (right panel). Herceptin^®^, a commercial product of anti-HER2 mAb trastuzumab, was used as a positive control for HER2 binding. For v23, both the initial product of MBP-scFv fusion protein and the final product of isolated scFv were examined. *K*_d_ values were determined using individual fitting curves. ND, not detected. (**B**) FACS results for monitoring target binding on SKOV3 (left panel) and MDA-MB-231 (right panel) cell surfaces. Raw data histograms are presented in Appendix A.

**Figure 5 biomolecules-13-01508-f005:**
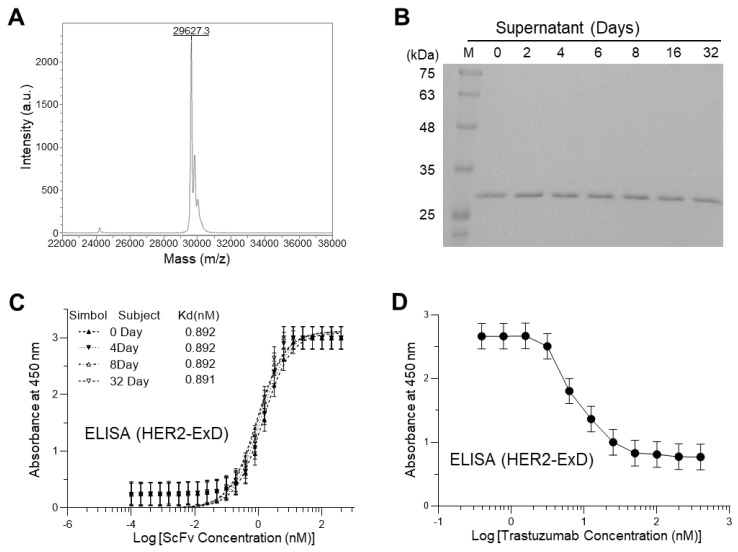
Identification and validation of the final product, v24 scFv. (**A**) MALDI-TOF mass spectrum measuring the absolute molecular mass. Sinapic acid was used as a matrix substance for measurement and IgG was used for linear-mode calibration of data. (**B**) Stability test by SDS-PAGE analysis. The scFv proteins were stored in a refrigerator (4 °C) for 32 days and sampled for analysis at indicated time points. (**C**) Functional stability tested by ELISA. The scFv proteins stored at 4 °C were sampled at indicated time points for measuring affinity to HER2-ExD. (**D**) ELISA results demonstrating a competitive antigen binding of the scFv with trastuzumab. 400 nM of hexahistidine-tagged scFv was initially bound to coated HER2-ExD (50 ng/mL), followed by dissociation upon increasing concentrations of trastuzumab added. Residual scFv was detected using anti-hexahistidine-HRP. Data in C and D are represented as mean ± SD for triplicate independent experiments.

**Table 1 biomolecules-13-01508-t001:** Protein productivity obtained from 1000 mg of wet cell pastes using the v21–v24 constructs.

Construct	Total Protein ^1^ (mg)	Final Product ^2^ (mg)
v21	217	0.143
v22	221	0.171
v23	286	2.142
v24	274	3.025

^1^ Quantification results for proteins in supernatants of cell lysates; ^2^ isolated scFv product collected from protein-L-affinity (v23) or nickel-affinity (v21, v22, and v24) chromatography.

## Data Availability

Raw data that support the findings of this study are available from the corresponding author (C.G.K.) upon reasonable request.

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
