# Peer review of "Development of an Anti-HER2 Single-Chain Variable Antibody Fragment Construct for High-Yield Soluble Expression in Escherichia coli and One-Step Chromatographic Purification"

_biomolecules, 2023, doi:10.3390/biom13101508_

Round 1
Reviewer 1 Report
The research manuscript by Byun et al. addresses two issues related to bacterial production of Anti-HER2 scFv: enhancing solubility and facilitating the process. While the authors used older, conventional techniques for scFv production and characterization, their proposed solution shows promise. The manuscript is written in simple language, making it readable to researchers in the field. However, there are some major concerns about the experiments that need addressing before the manuscript can be considered for publication.
Abstract:
The abstract is too lengthy (20 lines) and delves too deeply into methodology details.
Keyword:
"Maltose-binding protein (MBP)" is a relevant keyword for this study.
Introduction:
Lines 78-80: The authors introduced several approaches to promote soluble expression in E. coli. Another approach that should be considered is the "modification of media composition to obtain secretory proteins in E. coli" (http://dx.doi.org/10.1007/s12250-013-3286-9).
Materials and Methods:
This study requires a section on "Statistical analyses."
Results:
In line 256: The authors analyzed the ELISA and FACS results of v23 product and assumed that the fused MBP interfered with the steric interaction between scFv and the antigen. Is there any supporting claim or previously published article for this assumption?
In lines 260-262: This sentence is unclear. The authors discussed their observations based on a presumption without citing any supporting evidence.
Additionally, this discussion should be relocated to the "4. Discussion" section.
Figures 4 and 5:
How many repeats were performed for each independent experiment? Were they triplicates or duplicates? There is no statistical information provided.
Figure 5: The caption for Fig-C is unclear and insufficient. It might be beneficial to add a Fig-D for ELISA (HER-2).
Discussion:
In lines 316-321: The authors have already described steric interference by MBP. Therefore, there is no need to reiterate it in line 256 in the Results section.
The manuscript is written in simple language, making it readable to researchers in the field.
Reviewer 2 Report
The authors describe their work to study and improve the expression of an anti-HER2 scFv antibody in E. coli. Improving soluble and functional expression of scFvs is an important topic, and this work provides additional information to add to this challenging task. Overall, the work is done well and is explained clearly in the manuscript, but the following comments should be addressed before considering the work for publication:
1. There is no information regarding the cloning procedures, how the plasmids were modified to obtain the constructs in the manuscript.
2. In lines 134-136, the authors could give more details about the molar absorptivity values of constructs at 280 nm.
3. Table 1, if it is given in mg of product, that cannot be called a “yield”. Either the presentation or the annotation better be corrected.
4. In Line 236, there is no Lane L in Figure 2.
5. The authors should mention how the Kd values were calculated from ELISA.
6. In Figure 5, the authors may use A, B, C and D labels to follow easily.
7. What are the differences between construct v23 and the construct in your previous study (Ref 15)?
8. In Lines 351-353, the authors compare Kd values obtained from ELISA with the values from flow cytometry (Ref 36) and SPR (Ref 37). The authors could also mention these methods, since they don’t really correlate well and it may not be safe to compare them directly.
The paper does contain some minor grammar and spelling errors throughout that should be corrected. For example, in Line 16, Line 52, Line 64.
Line 129, the word “appropriate” is not necessary, and vague.
Round 2
Reviewer 1 Report
In the initial version of the manuscript, the primary concern revolved around the statistical analysis. In the revised version, the authors have addressed these concerns. The current version appears appropriate for publication; however, I recommend a thorough review of the English language in the main text to correct any typos or minor errors.
In the initial version of the manuscript, the primary concern revolved around the statistical analysis. In the revised version, the authors have addressed these concerns. The current version appears appropriate for publication; however, I recommend a thorough review of the English language in the main text to correct any typos or minor errors.